# The 50-nm Free Vesicles Visible in *Saccharomyces cerevisiae* Are Not COPII-Dependent

**DOI:** 10.3390/cimb47050336

**Published:** 2025-05-07

**Authors:** Alexander A. Mironov, Aurora Fusella, Galina V. Beznoussenko

**Affiliations:** 1Department of Cell Biology, IFOM ETS—The AIRC Institute of Molecular Oncology, Via Adamello, 16, 20139 Milan, Italy; galina.beznusenko@ifom.eu; 2Department of Neurosciences, Imaging and Clinical Sciences & Centre for Advanced Studies and Technology, University “G. d’Annunzio” of Chieti-Pescara, 66013 Chieti, Italy; aurora.fusella@gmail.com

**Keywords:** Golgi, coatomer, COPII, ER exit site, vesicles, intracellular transport

## Abstract

According to the current dogma, ER–Golgi transport is mediated by COPII-coated vesicles. However, numerous contradictions have emerged in this field. In this study, we demonstrate that *Saccharomyces cerevisiae* contains three distinct types of membrane spheres, with diameters of approximately 35–45 nm, 47–52 nm, and over 65 nm, respectively. The first type is Sso1-positive and primary associated with clathrin-positive endocytosis invaginations, which may function as exit sites for secretory soluble cargos. The second population is GOS1-positive and COPI-dependent. The third population represents secretory granules. Furthermore, we propose that several cornerstone studies supporting the COPII-vesicle model can have alternative interpretations. Our findings suggest that the predominant model of intracellular transport in *Saccharomyces cerevisiae* is the “kiss-and-run” mechanism.

## 1. Introduction

In the field of intracellular transport, the vesicular model (VM), cisterna maturation model (CM), and kiss-and-run model (KARM) compete for recognition as the prevailing paradigm. We have previously described the limitation of these models and proposed potential resolutions to the contradictions they present [1,2,3]. The vesicular model proposes that small membrane spheres (vesicles) form at endoplasmic reticulum (ER) exit sites (ERES). These vesicles detach from the proximal compartment and are moved toward the distal compartment. In most cases, vesicle formation is mediated by protein coat machineries such as coatomer and clathrin, although some vacuoles are formed by alternative, coat-independent mechanisms (reviewed by Mironov and Beznoussenko [3]). These vesicles are responsible for transporting cargo from ERES to the Golgi apparatus.

The cisterna maturation model (CM) proposes that entire compartments travel from ERES towards the Golgi, traverse the Golgi stack, and then proceed to the endosome or plasma membrane (PM). The core principle of the kiss-and-run model (KARM) is transient fusion between proximal and distal compartments [3]. Within the framework of the symmetrical kiss-and-run model, cargo diffuses through a narrow connection from the proximal to the distal compartment. In contrast, the asymmetrical kiss-and-run model suggests that fusion occurs between the edges of the proximal and distal compartments, while fission takes place within the proximal compartment. This fission separates the cargo-containing domain (connected to the rest of the proximal compartment via rows of pores or thin tubules) from the remaining proximal membrane. Following the fusion–fission event, the distal compartment increases in size, as the cargo-containing domain is now incorporated into it.

Although the superiority of the KARM has become increasingly evident in mammalian cells [4,5], this model continues to face strong objections in yeast. Critics point to the observation of so-called COPII-coated vesicles as evidence against KARM. Nevertheless, the view of intracellular transport remains largely confined to the vesicular model, often disregarding the substantial amount of evidence published in the last years.

The main cornerstone supporting the vesicular model of the ER–Golgi transport is the paper by Kaiser and Schekman [6]. We have previously discussed the limitations of this study and provided alternative interpretations in detail [3]. Kaiser and Schekman [6] demonstrated that, when the COPII machine was inhibited (specially by the blocking of Sec23 function) alongside inhibition of the membrane fusion machinery (via Sec17 or Sec18), 50-nm vesicles did not accumulate. However, vesicle accumulation was observed following inhibition of Sec22, an R-SNARE protein involved in ER–Golgi transport. Also, the methods applied by Kaiser and Schekman [6] for electron microscopy (EM)—specifically, the use of potassium permanganate to contrast the membranes in whole *S. cerevisiae* cells—did not provide the clearly defined vesicle contours.

Subsequently, Barlowe et al. [7], also from Schekman’s group, reported the isolation of so-called COPII-dependent 50-nm vesicles using purified COPII components and ER membranes. However, these authors did not provide evidence of Sec22 labeling on these vesicles. Notably, Sec22 interacts with Sec24 [8], Also, Rexach et al. [9] did detect Sec22 in “COPII vesicles”. Interestingly, the coats shown in Figures 7A and 8A,E of Barlowe et al. [7] appear morphologically similar to COPI coats (as described by Orci et al. [10]) and differ significantly from the COPII coats observed by Bannykh et al. [11], Bacia et al. [12], Bykov et al. [13], and Pyle et al. [14]. The conclusions presented by Barlowe et al. [7] were therefore inconsistent with our findings [15], in which we demonstrated that *S. cerevisiae* lacks COPII-coated buds in the ER [15].

There are other contradictions, namely normal function of Sec13p was not required for ER exit of the Hsp150 glycoprotein, whereas Sec23p and Sec31p and the GTP/GDP exchange factor Sec12p were required in functional form for the secretion of Hsp150 [16]. The ER exit of the soluble cargo Hsp150 glycoprotein did not require functional Sec24p [17].

Recently, Gomez-Navarro et al. [18] claimed to have identified COPII-dependent vesicles. For the identification of ERES during fluorescence, Gomez-Navarro et al. [18] and Melero et al. [19] used Sec23 and Sec16 tagged with fluorescent proteins. Their conclusion was based on the close proximity of 50-nm spheres to Sec23-positive areas as observed via light microscopy, a method that offers relatively low resolution. However, the authors did not demonstrate direct labeling of membrane buds in the ERES area, nor did they confirm the presence of a COPII coat on these structures. Thus, they did not conclusively prove that the membrane speres observed near ERES are really COPII-dependent.

Finally, just now using FIBSEM with rather low resolution, especially along the *z*-axis and automatic segmentation, which cannot resolve complicated issues of membrane continuity [5], Nair et al. [20] found COPI vesicles within ERES, although they did not prove that these 50-nm vesicles are COPI-dependent. Also, they did not claim that COPII vesicles exist.

However, the interpretation of vesicles with a diameter of 50 nm found in *S. cerevisiae* as vesicles that are the result of COPII function remained unchanged. This view has persisted despite more recent data on the size of COPII-dependent vesicles by Bannykh et al. [11], Matsuoka et al. [21], Bacia et al. [12], Beznoussenko et al. [15], Bykov et al. [13], and especially by Pyle et al. [14]. These findings forced us to reevaluate the long-standing conclusion regarding the identity and origin of these vesicles.

Our objective is to establish whether the 50-nm versicles at ERES are COPI- or COPII-dependent. Here, we have identified three distinct populations of membrane spheres, with size peaks around 40 nm, 50 nm, and 60–75 nm. Intermediate sizes are rare. Vesicles measuring 30–40 nm in diameter primarily originate from endocytic invaginations, as indicated by their Sso1 positivity. Importantly, we found no Sec22 in the 50-nm vesicles but did detect GOS1, suggesting that these vesicles are COPI-dependent.

## 2. Materials and Methods

*Saccharomyces cerevisiae* strains (BY4742, MMY2 [*MAT* a *ura3*], ura3-52, and his4-619 Mata) kindly provided by Drs. R. Kolling [Heinrich-Heine-Universitat Dusseldorf] and F. Kepes [22,23] were characterized previously [15]. Also, the strain *sec21-3* was provided by E. Gaynor and S. Emr (University of California, San Diego, CA, USA). The *sec21-1* (the temperature-sensitive [the nonpermissive temperature is equal to 37 °C] mutant) was provided by T. Kreis (Departement de Biologie Cellulaire, Universite de Geneve, 30 quai Ernest-Ansermet, CH-1211 Geneve 4, Switzerland). Yeast strains were grown exactly as described previously using cells grown to the mid-log phase [15]. Additionally, we used samples of cells transfected with GOS1-GFP that were prepared for our previous paper [15].

### 2.1. Antibodies and Plasmids

The rabbit polyclonal antibodies against Sec31p and Sec13 from *S. cerevisiae* (yC-20) were purchased from Santa Cruz Biotechnology (Heidelberg, Germany) and from Prof. V.V. Dolgikh (All-Russian Institute for Plant Protection, St. Petersburg, Russia). The rabbit polyclonal antibody against Sec21 was from Prof. D.C. Robinson [24,25]. A rabbit polyclonal antibody against Sec22 was from Biorbyt (Cambridge, Cambridgeshire CB5 8LA, UK; catalogue number: orb851130) and from Prof. V.V. Dolgikh. Rabbit polyclonal antibodies and antiserum against Sec22(Sly2)p were from R. Ossig and R. Grabowski [26,27,28]. A mouse monoclonal anti-SEC22 antibody was from Creative Biolabs (Industriepark Höchst Gebäude G830, 65929 Frankfurt am Main, Germany; Catalogue number: CBMOAB-03858CR). Rabbit antisera raised against recombinant GST-Sso1 was from Cocalico Biologicals (1683 Kramer Mill Road, Denver, PA, USA). Rabbit polyclonal antibodies against Sso1 were gifts from Profs. P. Brennwald [29] and E. Kuismanen [30]. For clathrin heavy chain (Chp1), the anti-clathrin light chain MAbs X-16 was obtained from the following sources: old aliquoted as kind gifts by Prof. F. Brodsky (University of California, San Francisco, CA, USA) from Thermo Fisher Scientific; 168 Third Avenue, Waltham, MA USA 02451 (catalog number PAS-144543) and from Santa Cruz (10410 Finnell Street; Dallas, Texas 75220; USA). Antibody against the clathrin heavy chain/CLTC (TD.1) catalog number sc-12734). Rabbit polyclonal antibody against GOS1 from *Saccharomyces cerevisiae* was from CUSABIO (7505 Fannin St., Ste 610, Houston, TX, 77054, USA) (Code CSB-PA009677XA01SVG). In order to obtain the signal, we used a high (3-fold higher than the standard) concentration of antibody and 3-fold high concentration of blocking agents (cold fish gelatin and Protein C from Sigma-Aldrich, St. Louis, MO, USA).

The plasmid expressing GFP-tagged Gos1p was characterized previously [15]. Yeast cells (strain BY4742) were transformed with these constructs using standard protocols. For transformed cells, we used polyclonal anti-GFP antibody (ab6556) from Abcam (Cambridge, UK). The plasmid containing pEGFP Sec22b was a gift from Dr. T. Galli (French Institute of Health and Medical Research; Addgene plasmid #101918; http://n2t.net/addgene:101918 (URL (accessed on 29 April 2025); RRID:Addgene_101918) and from Addgene (Catalog number 101918) [31].

In order to make the penetration of NEM into cells easier, we treated cells with U lyticase (or zymolyase) for dissolution and elimination of the yeast wall. It included β-1,3-glucan laminaritetraose-hydrolase and β-1,3-glucanase activities [32]. Finally, in order to increase the permeability of *S. cerevisiae* for Brefeldin A (BFA), we used the protocol developed by Pannunzio et al. [33] exactly as it was described by them.

### 2.2. Electron Microscopy

Routine EM, quick freezing, Tokuyasu cryosections, and EM tomography were performed as described [5,15,34,35]. Also, we used samples preserved for the previous paper [15], because labeling for GFP is higher than for the original antigens. We made cryosections with the thickness equal to 45–50 nm. In yeast, membranes are not always well visible [36]. Therefore, tomograms were made of these sections. The EM method of membrane visualization was improved. Initially, we obtained the 45–50-nm cryosections. Then, these sections were analyzed under EM, and when the zone of specific interest was visible, the EM tomographic analysis of this zone was performed at higher magnification. Further, 8 or 20 (usually up to 1/3 of the tomo-slices) centrally located serial tomography slices in each tomogram were subjected to averaging and merging them into one image using Photoshop.

### 2.3. Quantitation

We randomly selected a mitochondrion in the same cell or interest (previously selected using a random approach) or in a nearby cell that was at the same stage of development as the cell in which we measured the density of gold labeling on the vesicles. Then, a square morphometric grid was superimposed on the cryosection image, and the number of intersections of the grid with the contour of the membrane (I) was calculated, on which the labeling density and the number of gold granules in this measured area of the membrane (N) were measured. Then, the labeling density on this organelle membrane (No/Io) was divided by the labeling density on the inner mitochondrial membrane (IMM; Nm/Im), which was selected as an indicator of the background and was equal to No × Im/Io × Nm. Our improvements in the cryosection imaging method allowed us to analyze the distribution of vesicles in *S. cerevisiae*.

Labeling density (LD) of gold over the inner mitochondria membrane (IMM), ER, ERES, GC, PM, and vesicles with a diameter of 40 nm, vesicles with their diameters equal to 50 nm were estimated. The LD over IMM was taken as the background. Then, the ratio between LD in the above-mentioned organelles and LD in the IMM was estimated for each of two randomly selected cells. The mean of the two numbers was calculated and used as a statistical unit.

### 2.4. Sampling and Statistical Analysis

It is difficult to achieve the same conditions even within the same sample. Therefore, in many cases, we used random phenotypic analysis. Using three consecutive serial sections, we considered a round profile as a section of vesicles only in the case where this profile was visible only in the middle section. It is impossible to ensure the same incubation and processing conditions for even two cells. Phenotypical analysis was based on random sampling and the following probability: if there is a couple of experiments and, in both cases, one result, then the probability of this is 25%; if there are three coincidences, then the probability of this is 12.5%, 4–6.3%, 5–3.2%, and 6–1.6%. In each dish, we randomly selected three section and performed measurements. Usually, 6 statistical units were used. The mean estimated on the basis of these measurements was considered as a statistical unit, the number of which was equal to 6 in each experiment [37].

## 3. Results

Initially, we compared the main models of ER–Golgi transport and identified the key factors necessary to ensure proper functioning of this transport step (see Table 1). In our previous work [15], we demonstrated that Golgi-resident proteins are depleted in 50-nm vesicles, which are COPI-dependent at the level of the Golgi. We also addressed the primary objection to the KARM model by showing that different Golgi compartments can establish transient connections. Furthermore, we found that the ER itself does not display membrane buds on its surface, whereas ERES can form such a structure. The cisternal maturation model (CM) does not operate effectively at the ER–Golgi level, as 50-nm vesicles lack Golgi-resident proteins [15]. In the present study, we further investigated whether ERES generate membrane buds with diameters around 50 nm. Figure 1B–E,G present models of ERES as three-dimensional polygonal tubular networks (blue structures). The ER is shown in green (Figure 1B) and orange (Figure 1C,E). Within the ERES, clearly identifiable membrane buds are visible (indicated by red arrows), with sizes corresponding to nearby membrane networks approximately 50 nm and coats (Figure 1D,F: red and white arrows) resembling the COPI coat morphology described by Bykov et al. [13]. Their observation demonstrated that the ERES tubular network not only generates 50-nm buds but also maintains continuity with the ER (Figure 1E, domain shown in aqua color). Furthermore, free 50-nm vesicles were observed in proximity to ERES.

### 3.1. Vesicles in S. cerevisiae

To identify free vesicles, we improved the imaging quality of the cryosections. By applying a new method for membrane detection, we confirmed that the Golgi structure in *S. cerevisiae* remains consistent with our previous observations [15]. Structures associated with the secretory pathway in *S. cerevisiae* are presented (the panels of slices with free vesicles) in Figure 2. Notably, neither imaging approach revealed the presence of a coat structure on these free vesicles. In fact, no visible coats were detected on the round profiles (RPs) (Figure 1I–O). The distribution of vesicle diameters in *S. cerevisiae* is illustrated in Figure 1H, demonstrating three distinct peaks corresponding to average diameters of approximately 40 nm, 50 nm, and 65 nm (or larger). Among these, 50-nm RPs were predominantly located near ERES, with only occasional random cases observed in the cytosol.

Round profiles (RPs) with the peak of their diameters equal to 38–43 nm were predominantly observed near endocytic invaginations of the plasma membrane (PM), consistent with previous observations by Kukulski et al. [38,39]. In contrast, RPs with diameters equal to 60–80 nm were found near *trans*-Golgi, particularly in association with nodular structures and perforations, where the nodules exhibited similar cross-sectional diameters (Figure 1O). Studies by Rambourg’s group [22,23,40] demonstrated that diameters of these large vesicles correspond to those of the nodes of the *trans*-Golgi compartments.

However, in our images, where membrane visibility was enhanced, secretory granules with diameters of 65–75 nm were also detected near endocytic invaginations. Quantitative analysis showed that the relative numerical density of endocytic invaginations in the main cell body (when a bud was present) was 97 ± 19% of that in non-budding cells. In the budding compartment itself, the density increased to 222 ± 23% (N = 6).

Additionally, we performed a random analysis of cryosections to assess the presence of 40-nm and 50-nm vesicles. Across 10 central sections of *S. cerevisiae* cells, we identified only two vesicles with a diameter of 52 nm and one with a diameter of 42 nm, indicating that such vesicles are relatively rare.

The detection of two populations of vesicles near PM invaginations (one measuring 35–45 nm and other exceeding 65 nm) may indicate their multiple origins. While one possibility is that both types are derived from endocytic invaginations (Figure 1I,J), evidence from Kukulski et al. [36,37] argued against this for the larger vesicles. Instead, these larger vesicles can be attributed to the fragmentation of *trans*-Golgi disks with nodes [40,41,42].

### 3.2. Localization of COPII and COPI Markers on Endomembranes

Next, we examined whether ERES contain COPII and COPI molecular machines, which are important for the formation of vesicles. Labeling for Sec13 (a component of the COPII complex) was detected on part of the ER adjacent to ERES, as well as on the ERES themselves (Figure 3E,G,H). Sec13 was also observed on the ER (Figure 3E,G) (but not on the part of the ER situated near PM). Labelling for Sec22 (a SNARE protein) was localized on ERES and *cis*-Golgi. (Figure 3A,I,K). Sec31 showed similar distribution patterns (our unpublished observations). Labeling for Sec21 was observed mostly over Golgi and ERES (Figure 3B–D,F; Table 2). Notably, Sec21 labeling was also detected on both ERES and on the *cis*-Golgi and on the *medial*-Golgi but less frequently. Importantly, all round profiles (RPs) interpreted as free vesicles were negative for Se13, Sec21, and Sec22 but positive for GOS1 (Figure 3L). The labeling densities of different markers are presented in Table 2.

### 3.3. Inhibition of Sec23 and Its Effects

To test the COPII dependence of the 50-nm vesicles, we used the temperature-sensitive (*ts*) mutant of Sec23. If these vesicles are indeed COPII-dependent, then shifting the mutant cells to a restrictive temperature (37 °C) should result in the disappearance of the Golgi apparatus. Upon returning to the permissive temperature, the reactivation of COPII function (via Sec23 activity) should lead to the reappearance of COPPII-dependent vesicles, at least transiently. Furthermore, if membrane fusion is blocked during this recovery phase (e.g., using N-ethylmaleimide; NEM), these vesicles would be expected to accumulate.

We examined vesicle formation by shifting *ts*Sec23 mutant cells to 37 °C, followed by a return to 17 °C. According to the vesicular model (VM), restoration of Sec23 function should promote the formation of COPII-dependent vesicles. In our experiments, incubation at 37 °C led to the disappearance of Golgi cisternae. At the same time, the ERES tubular network accumulated, showing a 1.7- ± 0.1-fold (N = 6) increase in volume. Also, the total volume of the ER measured using a discretized rotator [37] increased 1.6- ± 0.2-fold (Figure 4A–C).

Previous studies by Rambourg et al. [40] and Morin-Ganet et al. [22,23] demonstrated that incubating yeast cells at 37 °C for 10 min led Golgi disassembly. In our experiments, we also incubated *ts*Sec23 mutant yeast cells at 37 °C for 10 min and then examined them. Under these conditions, the Golgi compartments disappeared. Simultaneously, numerous ER cisternae accumulated (Figure 4A–C), and the tubular networks became well developed (Figure 4D,E). After the heat treatment at 37 °C, Golgi compartments were undetectable in 81 ± 8% of the yeast cells.

Next, the cells were placed at 17–24 °C for 10 min and examined with EM. When the cells were incubated at 17 °C for 10 min, Golgi stacks containing multiple cisternae reappeared. Restoration of a normal temperature led to a reduction in the extent of tubular networks. Notably, stacks consisting of three and even more cisternae were visible, and in 30 ± 9% of cells, both a cisternal stack and adjacent ERES were observed (Figure 4F–H). This transformation of ERES into Golgi stacks has previously been described by Deitz et al. ([43]; see their Figure 3).

We then tested whether the 50-nm vesicles formed during recovery could be extremely short-lived. To address this, we used N-ethylmaleimide (NEM) to block membrane fusion. Notably, the microinjection of antibodies and other rapid molecular tools for *S. cerevisiae* practically do not work. Furthermore, the precise mechanism of action of temperature-sensitive mutants such as *ts*Sec23, *ts*Sec22, *ts*Se21, tsSec17, and tsSec18 remains unclear. For instance, heating a mutant may block its interaction with other proteins, but the exact targets remain unidentified. For example, heating tsSec17 or tsSec18 mutants can block interactions with the complete SNARE complex or impair SNARE sorting into distinct compartments of the secretory pathway. In contrast, the mechanism of NEM is well understood and widely used to inhibit membrane fusion events.

We used NEM as a well-characterized, although not very specific but fast, agent capable of rapidly blocking membrane fusion [44]. When this agent is used properly, the results are valid and reproducible. Moreover, other molecular tools were unable to block the fusion of vesicles quickly enough during the recovery of *ts*Sec23 function. To verify the efficacy of NEM, we examined the treated cells and observed an accumulation of electron-dense dots (Figure 4I), similar to those previously described by Kaiser and Schekman [6] and Morin-Ganet et al. [23]. In wild-type yeast cells, NEM treatment induced an accumulation of vesicular aggregates (Figure 4J), with contained vesicles with diameters roughly equal to 40 nm, 50 nm, and larger round profiles (more than 65 nm). Immunogold labeling revealed that the 40-nm vesicles were Sso1-positive, while the 50-nm vesicles were GOS1-positive but Sec22-negative (Figure 4J–N). These data align with our previous observations [15]. Notably, the absence of labeling for sec22 does not support the vesicular model (VM) but provides strong evidence in favor of the KARM. On the other hand, labeling for GOS1 on the 50-nm vesicles indicates that they are likely COPI-dependent.

To minimize the toxic effect of NEM, we examined cells for 3 min after NEM washout. After the treatment of cells with NEM, clusters of vesicles were observed, composed of 35–40 nm, 50 nm, and 65–70 nm vesicles, and a few larger vacuoles were visible. Small 35–40-nm vesicles were positive for Sso1, while the 50 nm vesicles were positive for GOS1, the analog of GS27/membrin, which is a marker of COPI-dependent vesicles [45,46], whereas the large vesicles were mostly negative for these markers.

To determinate whether 50-nm vesicles are formed during recovery from a COPII block, we assessed their formation after temperature normalization in the presence of NEM. We heated cells with the temperature-sensitive Sec23 mutation at 37 °C for 10 min and then placed the cells in NEM on ice for 20 min and then shifted to 17 °C. Under these conditions, we did not observe an accumulation of 50-nm vesicles in all six pairs of experiments (control versus experiment; the probability of mistake is equal to 1.6%).

There are several disadvantages of using the *erg6* mutation to obtain BFA sensitivity. The mutation itself causes a marked increase in permeability to sodium and lithium ions [47]. The efficiency of genetic transformation is lowered dramatically. Sexual conjugation is also greatly reduced. Also, the transport of tryptophan is lowered substantially [48]. The transport and processing of amino acid permeases, as well as some parts of secretory transport, are affected in these mutants [49]. Therefore, we used the protocol proposed by Pannunzio et al. [32]. In the presence of BFA, the 50-nm vesicles disappeared when used exactly as was described by Rambourg et al. [50]. As a result, we could not find an accumulation of 50-nm vesicles under the action of NEM in the presence of BFA in six independent pairs of experiments (the probability of a mistake is below 1.6%).

Next, we treated the cells with 1-µM NEM on ice and subsequently with dithiothreitol (DTT) to neutralize the NEM. During this process, we counted the number of 50-nm vesicles formed within the yeast cells over a period of three minutes. Aggregates containing both 50-nm and 35-nm vesicles were detected in 83 ± 5% of cells treated with NEM. Similar vesicle clusters have previously been reported after SNARE inhibition [22,23]. In these aggregates, the 50-nm vesicles were positive for GOS1 and negative for Sec22, while the 35-nm vesicles were positive only for Sso1 (Figure 4J–N). These results served as a control for experiments involving *ts*Sec23 mutants in the presence of NEM and temperature shifts.

Next, we used BFA and FLI-06 after the treatment of cells with NEM. In the first case, BFA blocked the accumulation of 50-nm vesicles induced by NEM, and in the second case, the 50-nm vesicles were accumulated in both the control samples and samples treated with FLI-06, an inhibitor of COPII [51]. In the first case, the 50-nm vesicles were not accumulated (not shown). In the second case, the effect was exactly the same as after the application of NEM alone (not shown). This indicates once more that the 50-nm vesicles are COPI-dependent but not COPII-dependent.

Finally, in order to show that COPII and COPI could be present in the same membrane and the 50-nm vesicles are formed by COPI, we used different markers. Figure 4O shows double labeling for Sec31 (10-nm gold: black arrow) and Sec21 (15-nm gold), suggesting once more that COPII and COPI could be present on the same membrane. On the other hand, Figure 4P demonstrates a COPI-coated bud (arrow) on the Sec31-positive ERES. Previously, we demonstrated COPI-coated buds after quick freezing in Figure S2k,l presented by Beznoussenko et al. [15]. A COPI coat on the cryosections could be seen in figures presented by Zeuschner et al. [52] and Martinez-Menárguez et al. [53]. Therefore, a COPI coat can be easily identified. Finally, Figure 4Q visualizes the 50-nm vesicles (arrow) negative for Sec 22 near the ERES positive for Sec 22. Thus, the 50-nm vesicles are COPI-dependent.

In order to evaluate this effect quantitatively, we performed six experiments on different days. Each experiment was composed of control and experimental samples. Control cells were treated with NEM, washed, and then DMSO was added. The experimental sample was treated with NEM, washed, and then DMSO containing BFA was added. The final concentration of BFA was always 3 µg/mL. In three minutes, both samples were fixed and then prepared for transmission electron microscopy. Then, in each sample, we randomly selected one section in all 12 samples and assessed whether accumulation of the 50-nm vesicles was visible. In all six control sections, we detected an accumulation of vesicles. In all experimental sections, we did not observe accumulation of the 50-nm vesicles. A similar mode was applied for COPII inhibitor FLI-06. In all 12 sections, we observed accumulation of the 50-nm vesicles exactly as it is shown in Figure 4I. In both cases, the probability of a mistake was 1.6% (see Materials and Methods). Thus, the conclusion that the 50-nm vesicles are formed by COPI is statistically correct.

## 4. Discussion

Here, we tried to clarify how the current data refine or challenge earlier interpretations. In order to do this, we need to explain several details that may seem tedious. However, without this context, it is almost impossible to convince readers. Also, here, we did not use emp24∆, lst1∆, and emp24∆ lst1∆, because these trains are not necessary for visualization of the 50-nm vesicles. For instance, Figure 2 presented by Gomes-Navaro et al. [18] demonstrates that the use of emp24Δ sec13Δ cells could induce significant alterations of the structure of ERES. This is especially important when such delicate issue as 50-nm vesicles are studied. Moreover, the transfection of not only the proteins significant for their function mutations but also normal but tagged proteins could induce significant alterations of the normal structure of secretory organelles [15]. For instance, Figure 2 presented by Gomes-Navaro et al. [18] demonstrated that the use of emp24Δ sec13Δ cells could induce significant alterations of the structure of ERES. We used the most straightforward methods based on tsSec23 and antibodies applied on cryosections. Our study did not include the labeling of ER exit sites with Sec-13-GFP and Sec16-GFP, because our antibodies were sensitive enough for our purposes. On the other hand, the GFP strains (Sec13-GTP and Sec16-GTP) and fluorescence microscopy images were not used, because in yeast, light microscopy cannot resolve the 50-nm vesicles due to dense cytosol and insufficient resolution. In yeast, where the cytosol is extremely dense, membrane morphology falls below the diffraction limit such that detailed structural information can only be obtained by EM [52,54]. Also, the spatial resolution of immune electron microscopy is much higher than that of light microscopy. Moreover, the transfection of yeast with additional protein, even tagged with GFP, induced alterations of the secretory compartments. We described this in detail in our previous study [15]. Finally, Sec13 is involved in the function of nuclear pores and even in vacuoles [55,56,57].

Using improved electron microscopy imaging, we demonstrated that the distribution of vesicle diameters exhibited peaks, with the main diameters at approximately 38–40, 50–52 nm, and more than 65–75 nm. The first two populations were positive to Sso1 and GOS1, correspondingly. The third population corresponded to secretory granules. Notably, round profiles (RPs) of intermediate size were rare, suggesting that only a limited number of molecular machines are responsible for forming free vesicles. In general, vesicles are rare in *S. cerevisiae*, which confirms our previous data regarding the presence of membrane spheres (particularly secretory granules) within the cytoplasm of yeast, especially near endocytic invaginations of the PM. The 50-nm vesicles are COPI-dependent. Following this, we tried to reinterpret the existing literature data in light of our observations.

For instance, the careful measurement of round profiles (RPs) in videos and supplementary images presented by West et al. [58] revealed the presence of three distinct populations of vesicles: those with diameters of 35 nm, 53 nm, and 75 nm. Notably, West et al. [58] and Papanikou et al. [59] did not show COPI-coated buds on the yeast Golgi. Mueller and Branton [60] claimed that isolated and subsequently uncoated clathrin-dependent vesicles from *S. cerevisiae* had diameters about 43 nm. Similarly, Huang et al. [61] observed that clathrin-coated vesicles from wild-type *S. cerevisiae* had a diameter of approximately 40 nm.

In our study, we observed 35–40-nm vesicles in proximity to endocytic invaginations. These 35-nm RPs were positive for Sso1, consistent with the findings by Rossi et al. [62], who also detected Sso1 labeling on small 35-nm vesicles. According to Kukulski et al. [38,39,63], the diameters of RPs near plasma membrane invaginations range between 30 and 42 nm. Furthermore, in Figures 2D,F and 7A,C presented by Buser and Drubin [64], the diameters of RPs near these invaginations are equal to 37–40 nm. These vesicles contain endocytic markers and are dependent on the invagination process [64]. Idrissi et al. [65] found labeling for clathrin on the lateral surface of these endocytosis invaginations. According to Kukulski et al. [63], in yeast, clathrin modulates vesicle scission but not the morphology of the invagination itself. Detachment of the clathrin coat from the membrane may affect *trans*-membrane area asymmetry, potentially facilitating membrane fission [66]. Alternatively, clathrin can be necessary to prevent the fusion of SGs with invagination. For such a fusion to occur, the clathrin coat must be removed, which is archived by budding and vesicle formation. Clathrin is also required for the retention of Kex2p in the Golgi [67]. Finally, in the absence of clathrin, 35–45-nm vesicles are not formed [68].

The second population of vesicles, formed at ER exit sites (ERES) and the Golgi complex, originates from COPI-coated buds. These vesicles are labeled for GOS1 [15], are not labeled for Sec22, have a diameter of approximately 50 nm, and are formed with the participation of COPI. Several lines of evidence support the conclusion that these 50-nm vesicles are COPI-dependent: (1) Vesicle dimensions: vesicles attributed to COPII typically have larger diameters equal to 65 nm [11], 70 nm [21], 78–80 nm [12], and 71–78 nm [13]. (2) The correspondence of bud and vesicle sizes: the dimensions of buds formed on ERES and Golgi match those of the 50-nm vesicles, supporting their identity as derivatives of these sites. (3) GOS1 labeling: the presence of GOS1 on the membranes of these vesicles, previously demonstrated by Beznoussenko et al. [15] and confirmed in the current study, is consistent with COPI dependency. (4) Absence of Sec22: the lack of Sec22, a protein typically associated with COPII-derived vesicles, further argues against a COPII origin. (5) Presence of COPI at ERES: the localization of COPI at ERES allows the formation of these vesicles at these sites, reinforcing the possibility of COPI involvement. Together, these points strongly argue against a COPII origin and instead support a COPI-dependent mechanism for the formation of 50-nm vesicles.

A comparison of the main models of ER–Golgi transport is present in Table 2. Based on our findings, we conclude that, in *S. cerevisiae*, the 50-nm vesicles can be COPI-dependent. This undermines the second major objection to the kiss-and-run model (KARM). The first objection, namely the existence of separate Golgi compartments, was already addressed earlier [15]. More recently, using the high-speed and high-resolution confocal microscopy studies by Kurokawa et al. [69] demonstrated that the *cis*-Golgi in budding yeast (*S. cerevisiae)* approaches and contacts the ERES. Following this interaction, the COPII coat cage collapses, and the *cis*-Golgi captures cargo directly. These observations support the view that COPII vesicles play only a minimal role in ER–Golgi transport and further strengthen the case for the KARM [1,2,3].

The KARM also explains the observations showing that, in yeast, when cargo exit from the ER is blocked, the Golgi apparatus disappears [23,70]. It explains the concentration of albumin, VSVG, mega-cargoes, and regulated secretory proteins along the *cis*-to-*trans* direction of Golgi stacks. Fusion and fissions events lead to residual connectivity along the secretory pathway, which may explain why labeled lipids can diffuse through the Golgi even in aldehyde-fixed cells, suggesting the presence of structural continuities [71,72,73]. If such membrane connections are continuously formed and then broken, and if a sufficient number of these connections exist at any defined moment, there should be at least one direct path from the ER to the plasmalemma.

Since KARM aspires to become a new paradigm, it must account for all existing observations without exception. Therefore, it is essential to explain the contradictions between KARM and the cornerstone papers by Kaiser and Schekman [6] and Barlowe et al. [7]. Although we previously provided a brief explanation for how the results could be interpreted with the framework of the kiss-and-run model [3]. Here, we offer a more detailed analysis.

Before proposing an alternative explanation, let us first reconstruct the logic presented by the authors. It was established that that Sec23, a subunit of COPII, determines the function of COPII and plays a critical role in the formation of COPII-coated vesicles. Sec23 is localized at the initial point of the secretory pathway—specifically, at the ER or ER exit sites (ERES)—but not at the Golgi apparatus. When the Sec23 function was inhibited, the ER was enlarged, likely due to cargo accumulation. On the other hand, Sec17 or Sec18 are involved in the regulation of SNARE complex activity, with a mediates membrane fusion event. When the Sec17 (or Sec18) function is blocked, the vesicles can still form but are unable to fuse with Golgi membranes, resulting in their accumulation. When COPII and SNARE function were simultaneously blocked, the 50-nm vesicles were not accumulated within 60 min. Based on this observation, Kaiser and Schekman [6] concluded that COPII vesicles are formed by the COPII machine with the participation of Sec23. These vesicles are accumulated if their fusion with Golgi is blocked. If COPII function is blocked by the inhibition of Sec23, then COPII vesicles do not form, and there is no accumulation of vesicles.

However, an alternative explanation for these data is also possible. Suppose that there are no COPII vesicles. Then, the question arises, why do vesicles accumulate? The answer is that they are formed from the Golgi membranes, but then why, when Sec23 function is simultaneously blocked, do vesicles not accumulate? It is well known that, in the absence of cargo delivery to the Golgi, the Golgi disassembles in about 4 min [23,70]. However, for the inhibition of SNARE function to begin to act, all the SNAREs ready for function must be used. Only then do the αSNAP/Sec17 and NSF/Sec18 proteins begin to disassemble the SNARE complexes [74]. That is, while the effect of a Sec23 blockade requires minimal time to manifest, Sec17 or Sec18 require more time. Therefore, if both proteins, namely Sec17 and Sec23, are inhibited simultaneously by increasing the temperature, the effect of Sec23 inhibition and the disappearance of the Golgi occurs first, and only then does Sec17 act, but by this time, the Golgi is no longer available as a source of 50-nm vesicles. Indeed, a small but significant increase in the number of 50-nm vesicles under these conditions can be seen in Figure 3A presented by Barlowe et al. [7].

The microsomes used by the authors contained Sec21 (γCOP), as shown in Figure 4B presented by Barlowe et al. [7]. This indicates that the microsomes already included ER exit sites (ERES), since coatomer-I was not present on the pure ER membranes. Upon the addition of purified coatomer-II (COPII) subunits, coatomer-I (COPI) moved along the membrane of the remaining ERES and formed COPI-coated buds. Because ARF1GAP was absent in the incubation medium, these buds remained stable. Simultaneously, coatomer-II attached to the membranes of both the remaining ERES and ER, forming COPII-coated tubules. Following this incubation, the membranes were subjected to centrifugation. The ER membranes, being heavier, were pelleted, while the supernatant was further processed by gradient centrifugation to isolate the vesicle-containing fraction. According to Figure 4 by Barlowe et al. [7], this final fraction did not contain coatomer-I. It was then subjected to another round of centrifugation, and the resulting pellet was analyzed by electron microscopy. This vesicle fraction appeared as a mixture of large and small vesicles without visible coats (see Barlowe et al. 1994; p. 898 [7]). This can be explained by the fact that COPI-coated structures incubated with GTPγS or GTP (but without ARF1GAP) remain tightly associated with the membranes and are not released during centrifugation [75].

Next, it was not the vesicle fraction itself but rather the supernatant (filler liquid) that was passed through a gel filtration column with small pores (apparently around 60 nm in diameter). As described by Beznoussenko et al. [15], during this process, tubules coated with coatomer-II (with an approximate diameter of 45 nm) were disrupted and fragmented into vesicles with diameters of about 50 nm. This transformation aligns with the transmembrane area asymmetry hypothesis [66,76], which suggests that mechanical spatial constraints can lead to membrane structure scission.

Simultaneously, COPI-coated buds were also separated from their donor membranes during filtration. Notably, the authors did not present any biochemical data confirming the presence of Sec21 (γCOP) in the filtrate obtained after this gel filtration step. Subsequently, the filtrate was split into two parts, and pellets were formed via centrifugation. These pellets were then processed either by embedding in Epon or gelatin for cryosectioning. Based on our experience with electron microscopy, sections from material embedded in Epon are typically cut from the very bottom of the pellet, which is likely to be enriched in denser components such as COPI-coated vesicles. In contrast, preparation for cryo-ultramicrotomy requires deeper trimming, often resulting in the removal of the bottom-most material. This technical difference may explain why Figure 8E presented by Barlowe et al. [7] showed circular profiles with the coat resembling COPI, while their Figure 8F–H did not reveal any clear coating, despite the fact that such coats (formed by either COPI or COPII) should be readily visible on cryosections, as previously demonstrated [53]. Furthermore, since GTPγS was not used in the incubation, the coat coverage observed in Figure 8E presented by Barlowe et al. [7] was incomplete, which is also consistent with earlier observations.

Thus, we demonstrated that the interpretations of the data presented by Kaiser and Schekman [6], Barlowe et al. [7], and Gomez-Navarro et al. [18] were not the only possible ones. Why are the 50-nm vesicles not COPII-dependent? These vesicles are Sec22- negative, and their diameter does not correspond to the well-established diameter of COPII buds and possible COPII-dependent vesicles. There is no accumulation of vesicles under the consecutive action of BFA and NEM. There is no accumulation of 50-nm vesicles during release after heating with the *tsSes23* mutant and then the application of NEM. Alternative interpretations should be considered. For instance, the 50-nm vesicles reported by Gomez-Navarro et al. [18] may be COPI-dependent, especially given the consistent presence of COPI components in the vicinity of COPII at ER exit sites. Moreover, the presence of 50-nm buds and the detection of Sec21 (γ-COP) at ERES further support the notion that ERES are capable of producing COPI-dependent 5-nm vesicles. As such, we concluded that 50-nm COPII-dependent vesicles do not exist.

## Figures and Tables

**Figure 1 cimb-47-00336-f001:**
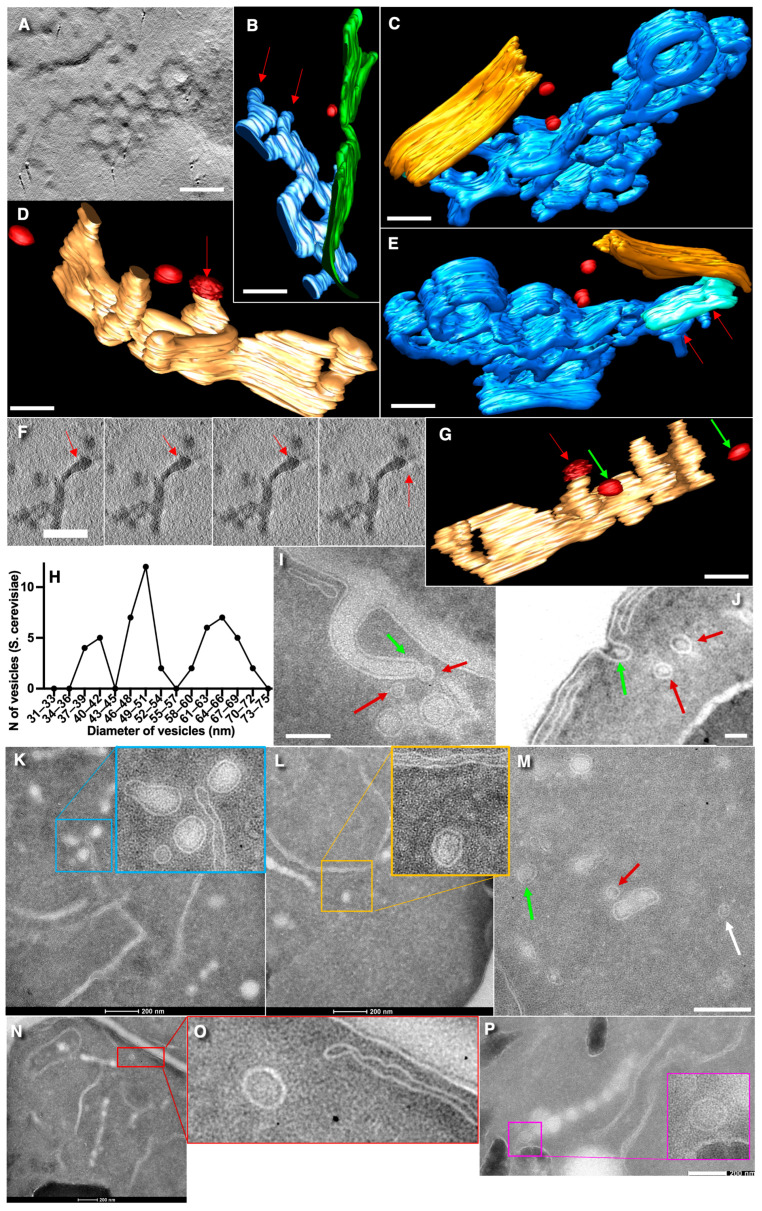
Structure of ER exit sites (ERES) and free vesicles in *S. cerevisiae.* (**A**) A single-electron tomography slice of ERES. (**B**–**E**,**G**) Three-dimensional models of tubular networks of ERES. (**B**) The ER is shown in green, ERES is blue, and the 50-nm vesicle is red. Red arrows indicate the 50-nm buds on the membrane of ERES. (**C**,**E**) Opposing three-dimensional view of the ERES connected to the ER. The ER is colored orange, while the membrane structure connecting the ER and ERES is shown in aqua. The ERES is blue, and the 50-nm free vesicles are red. The red arrows show the 50-nm buds. (**D**,**G**). Three-dimensional view (from opposite sides) of the ERES containing the COPI-coated bud (red arrow). Green arrows show the 52-nm vesicles. (**F**) Serial tomographic slices demonstrating a COPI-like coat (red arrows) on the ERES bud. This coat resembles the COPI coat described by Bykov et al. [13]. (**H**) Distribution of diameters of free vesicles in *S. cerevisiae* (three main peaks). (**I**–**O**) Representative cryosection images of various types of vesicles in *S. cerevisiae*. (**I**,**J**) Two 40-nm free vesicles (red arrows) located near an early endosomal invagination (green arrow). (**K**) Round profiles (RPs) with diameters of 40 nm and 80 nm. (**L**) The 54-nm RP. (**M**) RPs with diameters of 75 nm (green arrow), 49 nm (red arrow), and 42 nm (white arrow). (**N**) The 60-nm RP near the PM. (**O**) The 72-nm RPs (red arrow) near the *trans*-Golgi compartment. **Scale bars**: 300 nm (**A**,**F**); 250 nm (**B**,**C**,**E**); 150 nm (**C**,**E**); 100 nm (**D**,**G**); 40 nm (**I**); 200 nm (**J**–**L**,**P**); individual image scale bars are also shown.

**Figure 2 cimb-47-00336-f002:**
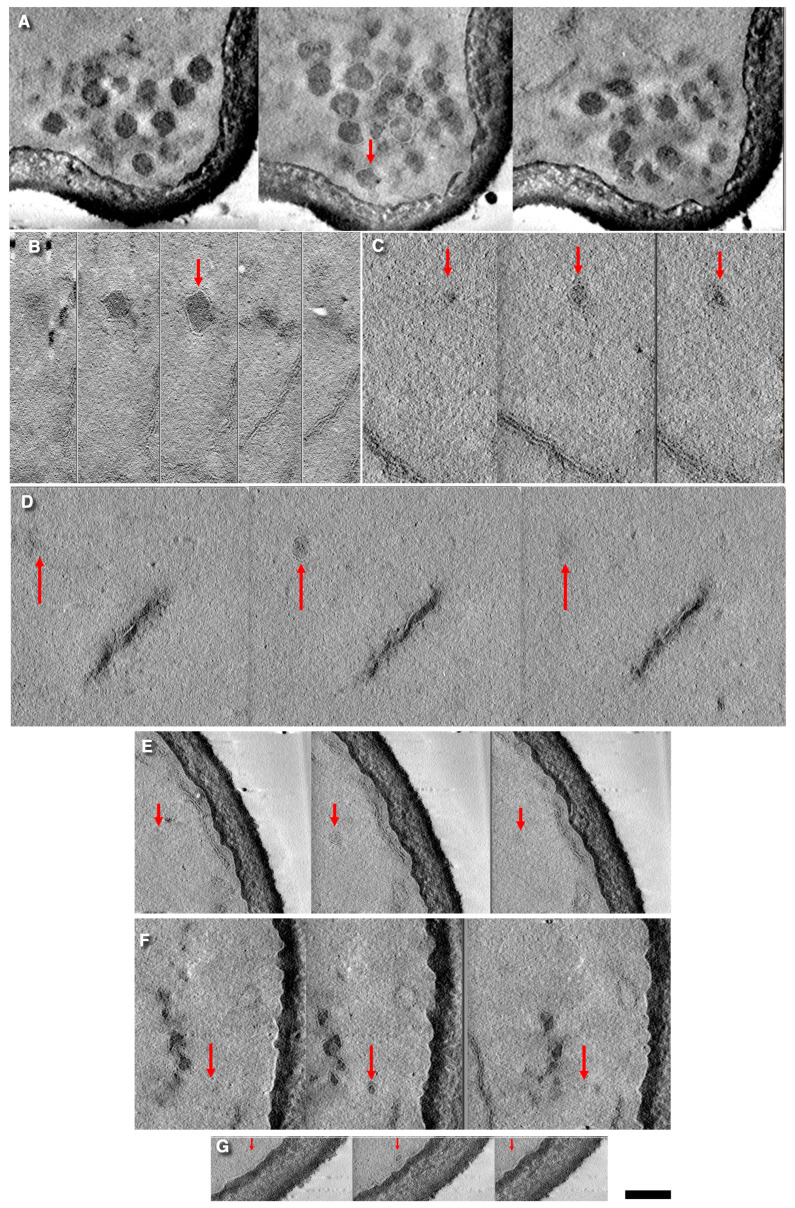
Representative free vesicles of different diameters observed in electron tomography datasets. Each image sequence shows the maximal diameter of RPs in a central tomographic slice (in the middle section, red arrow) and its absence in the previous image (to the left) and in the consecutive one (the right). Intermediate slices are avoided. (**A**) Fusion of a secretory granule (SG) with the plasma membrane (PM) of the cell bud (red arrow). (**B**) The 75-nm SG. (**C**–**E**) Examples of three vesicles with diameters around 50 nm. (**F**,**G**) Vesicles with diameters of approximately 30 nm. Red arrows show the positive of the vesicles. **Scale bars:** 140 nm (**A**,**D**); 120 nm (**B**); 170 nm (**E**,**G**); 205 nm (**F**).

**Figure 3 cimb-47-00336-f003:**
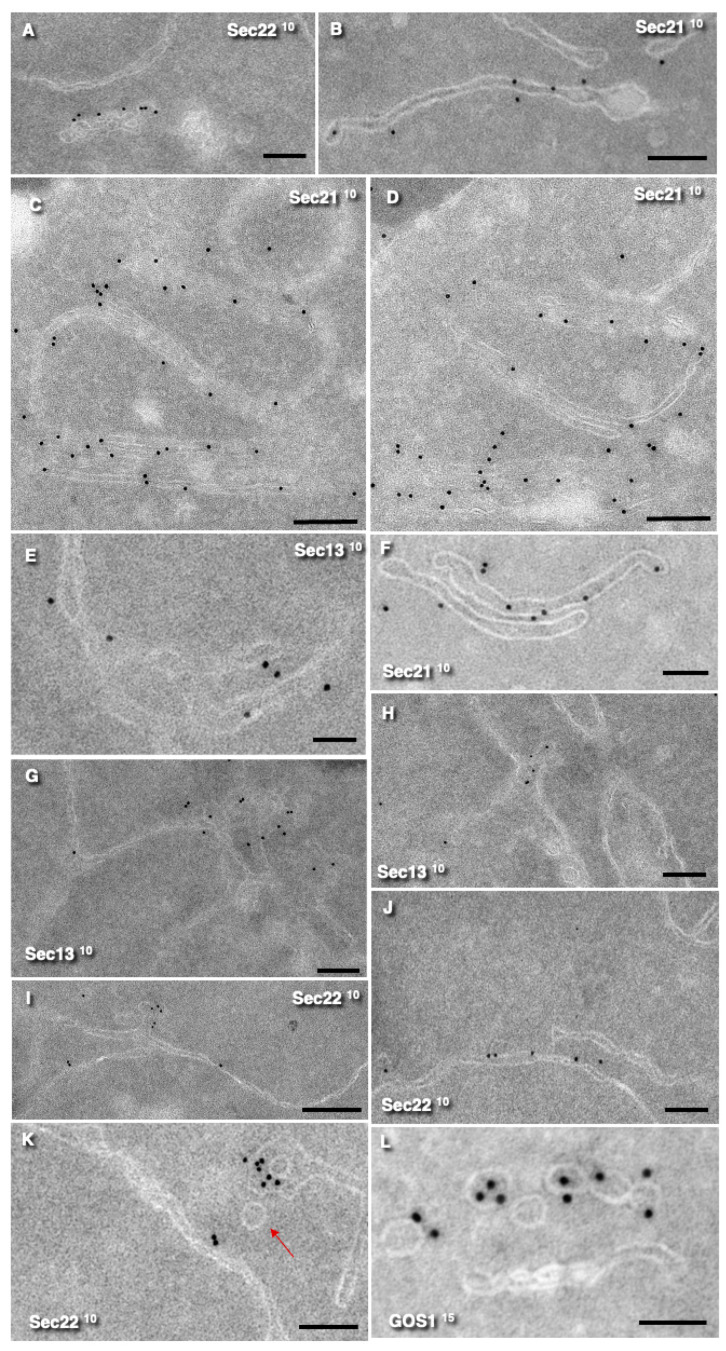
Immunogold labeling (10-nm and 15-nm gold) of ERES and Golgi compartments for Sec13, Sec21, Sec22, and GOS1 on ultrathin cryosections. The type of labeling is shown in images. (**A**,**I**–**K**) Labeling for Sec22. (**B**–**D**,**F**) Sec21 is present on ERES, *cis*-Golgi, and *medial*-Golgi. (**C**,**D**). Serial Tokuyasu ultrathin cryosections. (**E**,**G**,**H**) Sec13 is visible on the ER and ERES. (**L**) GOS1-positive 50-nm vesicles. Red arrow shows the 50-nm vesicle. **Scale bar:** 100 nm.

**Figure 4 cimb-47-00336-f004:**
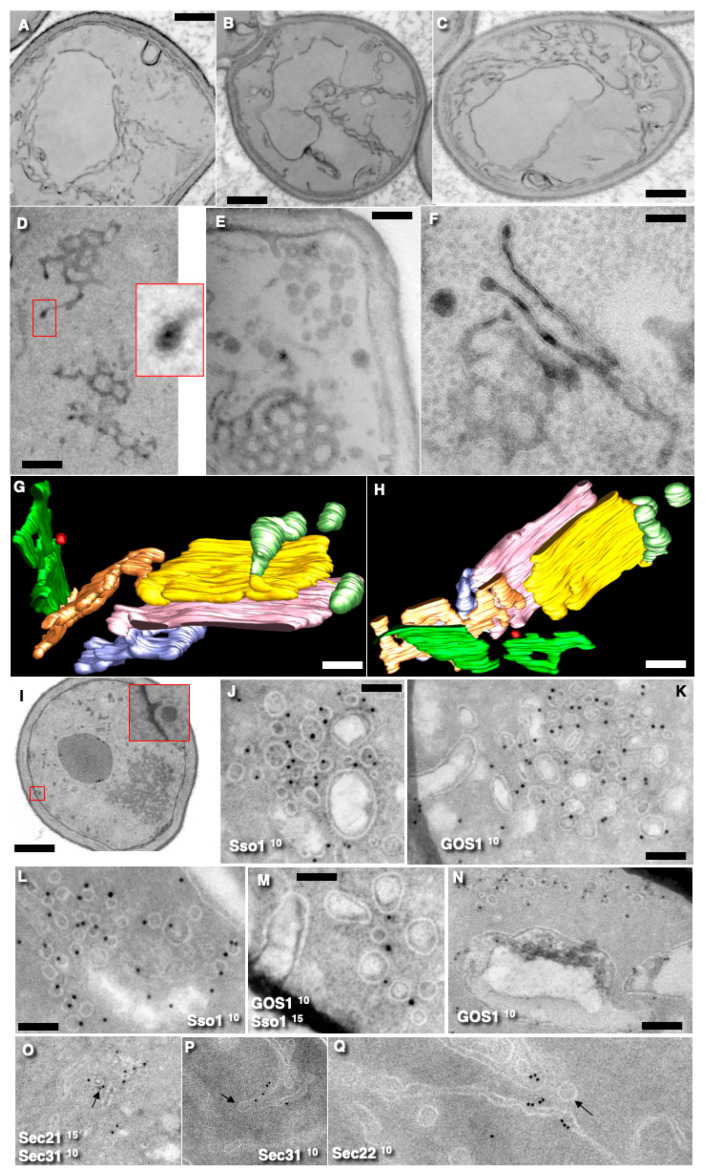
Alterations of the fine cell structure of *S. cerevisiae* containing the *ts*Sec23 mutation after a temperature shift and NEM treatment. (**A**–**C**) Heating cells at the restricted temperature (37 °C for 10 min) led to ER accumulation. (**D**,**E**) Accumulation of ERES in these heated cells. The red box and its magnified view show a COPI-coated bud. (**F**) The formation of large ERES and Golgi stacks after returning to the permissive temperature (17 °C) for 5 min. (**G**,**H**) Three-dimensional model of a Golgi stack formed during recovery composed of two medial cisternae. (**I**) The accumulation of RPs in the cell treated with NEM. The red box shows a secretory granule unable to fuse with the endocytic invagination. (**J**–**M**) Treatment of cells with NEM induced the accumulation of vesicles of three main size classes: 40 nm, 50 nm, and 65 and more nm. Immunogold labelling of vesicles within the vesicular aggregates for Sso1 and GOS1 after the blockage of membrane fusion with NEM. Markers and size of the gold are indicated on the images. (**J**–**L**) The 40–nm vesicles were positive for Sso1. (**M**) The 50-nm vesicles were positive for GOS1. (**N**) The 50-nm vesicles were GOS1-positive, whereas the 40-nm and the 65-nm round profiles were negative. (**O**) Double labeling for Sec31 (10-nm gold: black arrow) and Sec21 (15-nm gold). (**P**) COPI-coated bud (arrow) on the Sec31-positive ERES. (**Q**) The 50-nm vesicles (arrow) negative for Sec22 near the ERES positive for Sec22. **Scale bars:** 1130 nm (**A**–**C**,**I**); 350 nm (**D**,**E**); 175 nm (**F**–**H**); 70 nm (**M**); 85 nm (**J**,**L**); 140 nm (**K**); 200 nm (**N**).

**Table 1 cimb-47-00336-t001:** Requirements of models of intracellular transport.

Necessary Features	Model of Transport		
	VM	CM	KARM
1. COPII vesicles	necessary	not necessary	not necessary
2. Golgi resident proteins in COPI vesicles	not necessary	necessary	not necessary
3. Temporal fusion of the distal compartment with the proximal one	not necessary	not necessary	necessary
4. Presence of GOS1 in 50-nm vesicles	not necessary	not necessary	necessary
5. Presence of Sec22 in 50-nm vesicles	necessary	not necessary	not necessary

**Table 2 cimb-47-00336-t002:** Relative labeling densities (LDs in %) of different markers (the ratio related to LDs in the inner mitochondria membrane [IMM], which was considered as being equal to 100%; N = 6).

Membranes (N = 6)	Markers				
	Sec22	Sec21	GOS1	Sso1	Sec13
Vesicles 40 nm	111 ± 20	95 ± 18	98 ± 9	871 ± 80 *	125 ± 30
Vesicles 50 nm	93 ± 11	99 ± 10	101 ± 21	110 ± 12	133 ± 24
ER	344 ± 15 *	141 ± 26	121 ± 36	95 ± 24	232 ± 24 *
ERES	563 ± 34 *	710 ± 68 *	489 ± 46 *	89 ± 27	583 ± 35 *
Golgi	182 ± 41 *	637 ± 53 *	701 ± 27 *	119 ± 17	154 ± 56
PM	109 ± 19	120 ± 24	133 ± 25	1012 ± 134 *	126 ± 18

* Indicates data are significantly different from this parameter in IMM.

## Data Availability

Data are contained within this article.

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
