# Peer review of "The 50-nm Free Vesicles Visible in Saccharomyces cerevisiae Are Not COPII-Dependent"

_cimb, 2025, doi:10.3390/cimb47050336_

Round 1
Reviewer 1 Report
Comments and Suggestions for Authors
The study examines whether COPII vesicles mediate ER–Golgi transport in yeast. Using EM, immunogold labeling, and temperature-sensitive mutants, the authors claim that 50‐nm free vesicles are COPI-dependent. They further propose that the “kiss-and-run” model is a more accurate representation of intracellular transport in yeast than other competing models. However, there are many concerns:
- The manuscript is densely written, unpolished, and at times impenetrable:
- Line 31: A comma is missing before “in yeast.”
- Line 47: Remove the extra “and.”
- Line 72: Change “Sec 22” to “Sec22.”
- Line 105: A period is missing before “These.”
- Line 114: Delete “genes.”
- Line 156: “NEM” is not explained.
- Line 208: Change “COPI-dependent” to “COPI-dependent”.
- Line 266: Panel G is not labeled in Figure 1.
- Line 342: Revise the phrasing “although…but…” for clarity.
- Lines 343–344: Explain what is meant by “used cautiously.”
- Line 366: Explain “DTT” and delete the extra “)” character.
- Line 391: A scale bar is missing in Figure 4.
- Lines 399 and 446: “S. cerevisiae” should be italicized.
- Line 421: “SG” is not explained.
- Line 425: Change “COP1” to “COPI.”
- Line 442: Italicize “cis.”
- Lines 494 and 495: Revise “Since …, since…” for clarity.
- Lines 550 and 551: Correct the numbering problem.
- …
- The authors should rewrite the introduction section. It lacks a clear logical hierarchy and contains confusing expressions that need improvement. In particular, the last paragraph of the introduction reads as if it were thrown together.
- The discussion section is cluttered with references and technical details that distract from the central message. The authors should clarify how the current data improve or modify earlier interpretations.
- Remove homiletic content unrelated to the study—for example, line 540 (“Therefore, we urge everybody to start cooperating and conduct joint experiments, even with the participation of an independent third party.”). Instead, at the end of the paper, summarize the key findings and discuss the limitations of the study.
- The authors should also address whether the 50‑nm vesicles could be transitional intermediates or represent a subpopulation of COPII vesicles that lose coat components during isolation.
- Figure S1 has a confusing legend; for example, the “protein coat” is not shown in panel A, there are two different shades of green in panel B, and panel B does not match its legend.
See "Comments and Suggestions for Authors
"
Author Response
Reviewer 1
- The manuscript is densely written, unpolished, and at times impenetrable:
o Line 31: A comma is missing before “in yeast.”
o Line 47: Remove the extra “and.”
o Line 72: Change “Sec 22” to “Sec22.”
o Line 105: A period is missing before “These.”
o Line 114: Delete “genes.”
o Line 156: “NEM” is not explained.
o Line 208: Change “COPI-dependent” to “COPI-dependent”.
o Line 266: Panel G is not labelled in Figure 1.
o Line 342: Revise the phrasing “although…but…” for clarity.
o Lines 343–344: Explain what is meant by “used cautiously.”
o Line 366: Explain “DTT” and delete the extra “)” character.
o Line 391: A scale bar is missing in Figure 4.
o Lines 399 and 446: “S. cerevisiae” should be italicized.
o Line 421: “SG” is not explained.
o Line 425: Change “COP1” to “COPI.”
o Line 442: Italicize “cis.”
o Lines 494 and 495: Revise “Since …, since…” for clarity.
o Lines 550 and 551: Correct the numbering problem.
Our reply
Thank you very much for so excellent work. We corrected all these mistakes.
- The authors should rewrite the introduction section. It lacks a clear logical hierarchy and contains confusing expressions that need improvement. In particular, the last paragraph of the introduction reads as if it were thrown together.
Our reply
We corrected "Introduction", namely, we eliminated this part: For example, Koike and Jahn (2022) write in their abstract:
"...transport vesicles contain targeting signals such as Rab GTPases and polyphosphoinositides that are recognized by tethering factors in the cytoplasm and that connect the vesicles with their respective destination compartment". Similarly, Mogre et al. (2020) assert that a recycling pathway depends on the motor-driven transport of vesicles coated with coat protein I (COPI), which shuttle transport receptors and leaked ER-resident proteins back into the ER, thereby maintaining proteostasis within the organelles."
Also, we reorganized the logic of the whole text like in this fragment:
"However, an alternative explanation of these data is also possible. Let's assume that there are no COPII vesicles. Then the question arises, why is there an accumulation of vesicles? The answer is this. They are formed from Golgi membranes. But then why, when the Sec23 function is simultaneously blocked, vesicles do not accumulate, although the Golgi seems to be preserved. It is well known that in absence of cargo delivery to the Golgi, the Golgi rapidly disassembles, within approximately 4 minutes (Ayscough and Warren, 1994; Morin-Ganet et al., 2000). At the same time, in order for the inhibition of SNARE function to take effect, it is required that all ready-to-function SNAREs be consumed. That is, if it takes a minimum time for the Sec23 block effect to manifest, then it takes longer for Sec17 or Sec18 to manifest. Therefore, if both proteins, namely Sec17 and Sec23, are inhibited, then first the effect of Sec23 inhibition appears and Golgi disappears, and then Sec17 acts, but there is no longer the Golgi as a source of 50 nm vesicles. Indeed, a really small but noticeable increase in the number of 50-nm vesicles under these conditions can be seen in Figure 3A of the Barlowe et al. (1994) paper. Of interest, Kaiser and Schekman (1990) demonstrated that when Sec18 function is blocked and cargo synthesis is simultaneously inhibited with cycloheximide, vesicle accumulation does not occur presumably because the Golgi disappeared when synthesis of cargo is blocked.
- The discussion section is cluttered with references and technical details that distract from the central message. The authors should clarify how the current data improve or modify earlier interpretations.
Our reply
We corrected "Discussion" according to this demand. Also, we explained why it is necessary to stress details of the previous studies.
- Remove homiletic content unrelated to the study—for example, line 540 (“Therefore, we urge everybody to start cooperating and conduct joint experiments, even with the participation of an independent third party.”). Instead, at the end of the paper, summarize the key findings and discuss the limitations of the study.
Our reply
We eliminated this and added some kind of summary in the end of "Discussion".
- The authors should also address whether the 50 nm vesicles could be transitional intermediates or represent a subpopulation of COPII vesicles that lose coat components during isolation.
Our reply
The 50-nm COPII-dependent vesicles do not exist. We indicated this in the text.
- Figure S1 has a confusing legend; for example, the “protein coat” is not shown in panel A, there are two different shades of green in panel B, and panel B does not match its legend.
Our reply
We eliminated this figure.
Reviewer 2 Report
Comments and Suggestions for Authors
In the manuscript the authors try to prove that the 50-nm free vesicles visible in Saccharomyces cerevisiae are not dependent on COPII. They postulate that there are three distinct populations of membrane spheres with peaks in the region of 40, 50, and 60-75 nm and additionally they state that intermediate sizes are rare. The results observed are not obvious and are intriguing. Therefore, I consider it an interesting manuscript despite its flaws and experimental gaps. The authors themselves point out that the 50-nm vesicles are formed during cell recovery could have very short period of life and therefore they are difficult to research. Regardless, the authors should propose a different method of validating their findings in the article to strengthen the hypothesis. For example, the use of additional GFP strains (Sec13-GTP, sec16-GTP) and fluorescence microscopy images. Why didn't the authors use strains that would allow them to investigate the function of the inner COPII coat with three different strains such as emp24∆, lst1∆, and emp24∆ lst1∆? Authors should test their hypothesis in the presence of brefeldin A and/or inhibitor FLI-06.
Author Response
Reviewer 2.
- In the manuscript the authors try to prove that the 50-nm free vesicles visible in Saccharomyces cerevisiae are not dependent on COPII. They postulate that there are three distinct populations of membrane spheres with peaks in the region of 40, 50, and 60-75 nm and additionally they state that intermediate sizes are rare. The results observed are not obvious and are intriguing. Therefore, I consider it an interesting manuscript despite its flaws and experimental gaps.
Our reply
Thank you very much for your suggestions.
- The authors themselves point out that the 50-nm vesicles are formed during cell recovery could have very short period of life and therefore they are difficult to research.
Our reply
In order to avoid such kind of confusions we treated cells with NEM during the recovery of cells to normal temperature and did not find accumulation of the 50-nm vesicles.
- The authors should propose a different method of validating their findings in the article to strengthen the hypothesis. For example, the use of additional GFP strains (Sec13-GTP, sec16-GTP) and fluorescence microscopy images.
Our reply
Thanks a lot for your proposal. However, light microscopy especially in yeast cannot resolve the 50 nm vesicles due to low resolution.
- Why didn't the authors use strains that would allow them to investigate the function of the inner COPII coat with three different strains such as emp24∆, lst1∆, and emp24∆ lst1∆?
Our reply
These strains were already examined. Accumulation of the 50-nm vesicles were not detected. Therefore we used the most straight forward methos based on tsSec23.
- Authors should test their hypothesis in the presence of brefeldin A and/or inhibitor FLI-06.
Our reply
Thanks a lot again and again. We used FLI-06 and could not find any obvious effect due to problem with its penetration through the plasma membrane and cell wall. The effect of Brefeldin A was examined in details by Rambourg et al. (1995). Under the action of this drug the 50-nm vesicles disappeared. Thus, this drug is not very useful for our purposes.
Rambourg, A., Y. Clermont, C.L. Jackson, and F. Kepès. 1995a. Effects of brefeldin A on the three-dimensional structure of the Golgi apparatus in a sensitive strain of Saccharomyces cerevisiae. Anat. Rec. 241:1–9. https://doi.org/10.1002/ar.1092410102

Round 2
Reviewer 1 Report
Comments and Suggestions for Authors
The authors have addressed all of my concerns. I recommend acceptance.
Author Response
We improved the presentation of our data.

Reviewer 2 Report
Comments and Suggestions for Authors
The authors should clearly address all the issues raised in my previous review in their discussion. If they did not perform the studies I suggested, they should discuss these aspects in their discussion, explaining why, for example, they did not use GFP strains (Sec13-GTP, sec16-GTP) and fluorescence microscopy images, which have a short half-life of 50 nm, or why they did not use GFP strains (Sec13-GTP, sec16-GTP) and fluorescence microscopy images, which have a short half-life of 50 nm, they did not use GFP strains (Sec13-GTP, sec16-GTP) and fluorescence microscopy images, with a short half-life of 50 nm vesicles, why they did not use strains such as emp24∆, lst1∆, and emp24∆ lst1∆, and brefeldin A and/or inhibitor FLI-06.
Author Response
Reviewer 2
- The authors should clearly address all the issues raised in my previous review in their discussion.
Our reply
We addressed all issues. In our reply there is the following text: "3. The authors should propose a different method of validating their findings in the article to strengthen the hypothesis. For example, the use of additional GFP strains (Sec13-GTP, sec16-GTP) and fluorescence microscopy images. Our reply
Thanks a lot for your proposal. However, light microscopy especially in yeast cannot resolve the 50 nm vesicles due to low resolution."
- If they did not perform the studies I suggested, they should discuss these aspects in their discussion, explaining why, for example, they did not use GFP strains (Sec13-GTP, sec16-GTP) and fluorescence microscopy images, which have a short half-life of 50 nm.
Our reply
Sec13-GFP and Sec16-GFP are useful for immune fluorescence analysis which cannot resolve the 50-nm vesicles. Our study does not include the labelling of ER exit sites with Sec-13-GFP and Sec16-GFP because our antibody was sensitive enough for our purposes. Spatial resolution of immune electron microscopy is much higher than that of light microscopy. On the other hand, Sec13 is involved into function of nuclear pores and even vacuole (Ryan and Wente, 2002; Enninga et al., 2003; Dokudovskaya et al., 2011). Moreover transfection of yeast with additional protein even tagged with GFP induced alterations of the secretory compartments. We described this in details in our previous study (Beznoussenko et al., 2016). Also, we did not examine "a short half-life of 50 nm" because this issue is not included into our tasks.
- ... or why they did not use GFP strains (Sec13-GTP, sec16-GTP) and fluorescence microscopy images, which have a short half-life of 50 nm, they did not use GFP strains (Sec13-GTP, sec16-GTP) and fluorescence microscopy images, with a short half-life of 50 nm vesicles.
Our reply
This is a repetition of the previous text of this comment.
- Why they did not use strains such as emp24∆, lst1∆, and emp24∆ lst1∆,
Our reply
In our first-round reply sent to reviewers there is the following text: "4. Why didn't the authors use strains that would allow them to investigate the function of the inner COPII coat with three different strains such as emp24∆, lst1∆, and emp24∆ lst1∆? Our reply: These strains were already examined. Accumulation of the 50-nm vesicles were not detected. Therefore we used the most straight forward methos based on tsSec23."
Thus, all these trains are not necessary for the visualization of the 50-nm vesicles. We did not use emp24∆, lst1∆, and emp24∆ lst1∆ because these trains are not necessary for visualization of the 50-nm vesicles. For instance, Figure 2 presented by Gomes-Navaro et al. (2020) demonstrates that the use of the emp24Δ sec13Δ cells could induce significant alterations of the structure of ERES. This is especially important when so delicate issue as the 50-nm vesicles are studies. Moreover transfection of not only the proteins containing significant for their function mutations but also a normal but tagged proteins could induce significant alteration of normal structure of secretory organelles (Beznoussenko et al., 2016). For instance, Figure 2 presented by Gomes-Navaro et al. (2020) demonstrates that the use of the emp24Δ sec13Δ cells could induce significant alterations of the structure of ERES. This is especially important when so delicate issue as the 50-nm vesicles are studies. We used the most straight forward methods based on tsSec23 and antibodies applied on cryosections. Our study does not include the labelling of ER exit sites with Sec-13-GFP and Sec16-GFP because our antibodies were sensitive enough for our purposes. On the other hand, the GFP strains (Sec13-GTP, Sec16-GTP) and fluorescence microscopy images were not used because in yeast light microscopy cannot resolve the 50 nm vesicles due to dense cytosol and insufficient resolution. In yeast, where the cytosol is extremely dense, membrane morphology falls below the diffraction limit such that detailed structural information can only be obtained by EM (Orci et al., 1991; Zeuschner et al., 2006). Also, spatial resolution of immune electron microscopy is much higher than that of light microscopy. Moreover transfection of yeast with additional protein even tagged with GFP induced alterations of the secretory compartments. We described this in details in our previous study (Beznoussenko et al., 2016)". Finally, Sec13 is involved into function of nuclear pores and even in vacuole (Ryan and Wente, 2002; Enninga et al., 2003; Dokudovskaya et al., 2011).
If we would measure the colocalization of COPI and COPII subunits in S. cerevisiae at the level of light microscopy, then the resolution especially along Z axis will be low. So far, no one has used measurement of colocation between COPI and COPII in the village of S. cerevisiae because of its low accuracy. In Internet, we could not find images showing localization of Sec13 and Sec21 (gCOP) in the same cell. However, in order to satisfy the reviewer 2 we added the text explaining this into Discussion.
- Why they (we) did not use brefeldin A and/or inhibitor FLI-06.
Our reply
We used BFA and FLI-06 after treatment of cells with NEM. In the first case BFA blocked accumulation of the 50-nm vesicles induce by NEM and the second case, the 50-nm vesicles were accumulated in both control samples and samples treated with the FLI-06, an inhibitor of COPII (Denisova et al., 2021). This indicates once more that the 50-nm vesicles are COPI- but not COPII-dependent. We included this information into Methods, Results and Discussion parts.
